# GAUSSIANCLIN: MULTIMODAL FEATURED GAUSSIAN SPLATTING FOR DYNAMIC CLINICAL VIDEOS

## ABSTRACT

Reconstructing dynamic 3D models from clinical videos is crucial for medical applications such as surgical visualization, robot-assisted surgery, and medical training. However, the clinical environment presents unique challenges, including limited surface textures, inconsistent lighting, and the need for expert-level medical knowledge, making it difficult for non-experts to directly apply existing techniques. To address these challenges, we present *GaussianClin*, a novel approach that enhances 3D modeling capabilities in dynamic clinical videos by leveraging multimodal feature-based Gaussian splatting (GS). By embedding trained multimodal feature fields into the radiance field, *GaussianClin* integrates general medical knowledge and improves the performance of GS in tasks like 3D tissue visualization, real-time object enhancement, clinical instrument and organ segmentation, and medical visual question answering. To effectively capture temporal dynamics and tissue deformations, we further introduce a spatiotemporal graph distillation, which significantly improves handling deformable tissues compared to standard GS methods. Experimental results demonstrate that *GaussianClin* enables clinical 3D expert models to leverage massive pre-trained 2D multimodal foundation models, paving the way for advancements in robot-assisted surgery.

## 1 INTRODUCTION

High-quality reconstruction of surgical videos is becoming increasingly important in modern medicine. It not only provides precise anatomical information for surgical planning but also aids surgeons in making accurate decisions during real-time navigation and offers immersive, realistic virtual environments for medical training. In the field of surgical scene reconstruction, traditional methods such as structured light and SLAM (Fuentes-Pacheco et al., 2015; Kazerouni et al., 2022) have been widely used. However, these techniques often struggle with the complex geometries of soft tissues and the dynamic nature of surgical environments, leading to inaccuracies and incomplete reconstructions. While volumetric reconstruction approaches have shown potential, they still face challenges in capturing the full complexity of surgical scenes, particularly when dealing with deformable tissues and inconsistent lighting conditions. These limitations have spurred the development of more advanced reconstruction techniques that aim to meet the increasing demands for both precision and real-time performance.

In recent years, Neural Radiance Fields (NeRF) (Mildenhall et al., 2021; Gao et al., 2022), which model 3D scenes by learning the volumetric radiance and density from multiple 2D images, have emerged as a powerful technique for reconstructing surgical scenes. For example, EndoNeRF (Wang et al., 2022) represents dynamic scenes as canonical fields with a time-dependent displacement field, effectively capturing the subtle nuances of deformable tissues. However, NeRF-based methods require repeatedly querying the radiance field at multiple points and rays to render each view, which significantly limits their rendering speed and poses challenges for real-time intraoperative applications. To address these limitations, 4D Gaussian Splatting (4DGS) techniques have been widely adopted in surgical reconstruction. 4DGS represents the scene as an optimizable Gaussian model initialized from motion structure techniques. Recent advancements, such as Gaussian SLAM (Yugay et al., 2023), GS-SLAM (Yan et al., 2024), and SGS-SLAM (Li et al., 2024b), have extended 4DGS to optimize camera poses and propose SLAM strategies for rapidly reconstructing 3D scenes from 2D videos. Among these, various Gaussian-based methods, such as SurgicalGaussian (Xie et al., 2024), have shown particular promise for real-time reconstruction of deformable tissues in mini-

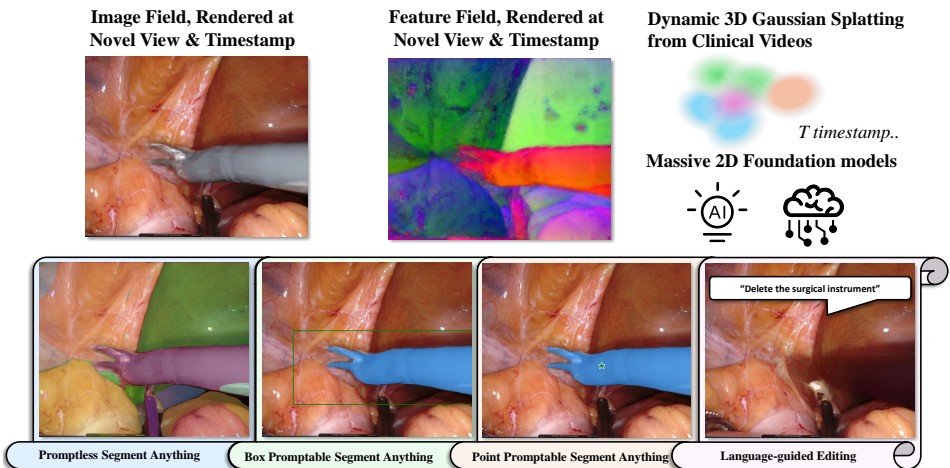

Figure 1: **Enhancing 3D Gaussian Splatting for Dynamic Surgical Scene Reconstruction.** The framework integrates multimodal feature fields from pre-trained 2D foundation models into 3D rendering, improving 3D Gaussian Splatting for clinical and surgical videos. The top row shows a rendered surgical image, feature field, and Gaussian splatting. The bottom row demonstrates promptless segmentation, promptable segmentation, and language-guided editing. These tools enhance tissue visualization and instrument manipulation for robot-assisted surgery.

mally invasive surgical environments. These technological advancements open up new possibilities for more precise and real-time surgical planning, navigation, and training, significantly enhancing the overall quality of medical procedures and education.

Despite significant advancements in surgical scene reconstruction techniques (Liu et al., 2024b; Yang et al., 2024), a key challenge persists in making the results both comprehensible and functional for practical medical use. Unlike the intuitive nature of natural scene reconstruction, medical reconstructions—while often technically successful—are difficult for non-professionals to interpret, as they require specialized medical expertise to understand the complex anatomical structures and dynamic surgical environments (Rodríguez et al., 2022; Liu et al., 2024a). For instance, interpreting tissue deformations caused by physiological processes or the interactions between surgical instruments and tissues requires a deep understanding of human anatomy and surgical procedures (Wang et al., 2022; Zha et al., 2023). This difficulty in interpretation is further compounded by the fact that current 4D Gaussian models struggle to support critical surgery-related downstream tasks, such as real-time scene editing, semantic understanding, and automatic surgical report generation, which are essential for improving surgical efficiency and decision-making (Mahmoud et al., 2017). As a result, these limitations not only hinder the accessibility of reconstruction outputs but also restrict broader potential in advancing surgical automation, preoperative planning, and postoperative evaluation.

Secondly, surgical environments typically present limited surface textures and inconsistent lighting conditions (Batlle et al., 2023; Yang et al., 2023), which further complicate the reconstruction process. Traditional reconstruction methods struggle to achieve sufficient rendering quality in areas with smooth tissue surfaces or complex reflections, affecting the overall accuracy and reliability of the reconstruction. This challenge is particularly pronounced in endoscopic surgeries, where surgeons frequently adjust camera positions to track specific tissues and improve visibility, making it difficult to capture precise camera trajectories. Additionally, the dynamic nature of surgical scenes—such as tissue deformations caused by physiological processes and interactions with surgical instruments—further increases the complexity of the reconstruction task (Liu et al., 2024b). These limitations highlight the urgent need for innovative approaches to improve reconstruction quality. Therefore, we propose a novel framework that not only enhances reconstruction quality but also improves the accessibility and applicability of these models across various clinical scenarios, particularly under low-light and complex surgical conditions. Such a framework will pave the way for advancements in robot-assisted surgery and medical image processing, while providing precise and reliable tools for preoperative planning, intraoperative navigation, and postoperative evaluation.

To address the above-mentioned challenges, we propose *GaussianClin*, an innovative framework which enhances the dynamic performance of Gaussian splatting by integrating multimodal feature fields from pre-trained 2D foundation models into the 3D rendering process. By embedding these trained feature fields into the Gaussian radiance field, *GaussianClin* injects semantically rich medical knowledge, enabling GS to perform tasks such as 3D tissue visualization, object enhancement, scene understanding, and language-guided interactions. Given the gap between 2D foundation models and 3D Gaussian radiance fields, further widened by the temporal dynamics and tissue deformations in clinical videos, we introduce a spatiotemporal graph distillation mechanism to effectively embed these features into the dynamic field. Experimental results demonstrate that *GaussianClin* enables clinical 3D expert models to leverage large-scale pre-trained 2D multimodal foundation models and improves rendering quality in clinical scenarios, especially under challenging conditions like low lighting and occlusion, paving the way for advancements in robot-assisted surgery. In summary, our key contributions are as follows:

- We introduce *GaussianClin*, which enhances dynamic Gaussian Splatting by embedding foundation model feature fields, addressing challenges in rendering quality and semantic understanding in surgical scene reconstruction.

- We propose a spatiotemporal graph distillation mechanism that better captures temporal dynamics and tissue deformations, improving the handling of deformable tissues compared to traditional GS methods.

- We demonstrate that *GaussianClin* supports real-time tasks such as rapid segmentation, language-guided editing, and medical visual question answering, expanding its utility in surgical applications.

- Our method achieves real-time rendering with high image quality, significantly accelerating both training and rendering speeds, enabling advanced intraoperative applications in robot-assisted minimally invasive surgery.

## 2 RELATED WORK

### 2.1 3D SCENE RECONSTRUCTION FROM STATIC SURGICAL VIDEOS

Reconstructing 3D scenes from 2D images is crucial in many fields, especially in surgical settings. Traditional methods like Structure-from-Motion (SfM), as in COLMAP (Schönberger & Frahm, 2016), and SLAM-based approaches (Zhou & Jagadeesan, 2019; Song et al., 2017; Zhou & Jayender, 2021) have been used in endoscopic reconstruction, utilizing depth and color data to create 3D point clouds. However, these methods often struggle in dynamic surgical environments due to their assumption of scene stability. Recent innovations, such as Gaussian Splatting (Kerbl et al., 2023; Bao et al., 2024), represent scenes as optimizable Gaussian entities initialized from SfM data, offering faster and more accurate 3D reconstructions. Extensions like Gaussian-SLAM (Yugay et al., 2023), GS-SLAM (Yan et al., 2024), and SGS-SLAM (Li et al., 2024b) further enhance these methods, while specialized adaptations like EndoGSLAM (Wang et al., 2024) address challenges such as reflections, occlusions, and tissue deformations. These advancements are paving the way for more precise 3D reconstructions in surgical applications, potentially transforming surgical planning and intraoperative navigation.

### 2.2 4D SCENE RECONSTRUCTION FROM DYNAMIC SURGICAL VIDEOS

Reconstructing dynamic surgical scenes is more complex than static ones due to tissue deformations from physiological processes (e.g., respiration, heartbeat) and interactions with surgical instruments, which complicate camera pose estimation. Methods like RoDyNeRF (Liu et al., 2023) optimize static and dynamic radiance fields, while Free SurGS (Guo et al., 2024) improves pose estimation by minimizing projection and optical flow loss. Despite progress, real-time performance and precision remain challenging, which is critical for robot-assisted minimally invasive surgery, supporting 3D models for preoperative planning, AR/VR training, and intraoperative guidance. NeRF-based models, such as EndoNeRF (Wang et al., 2022), capture deformations through time-variant neural displacement fields. Innovations like HexPlane and 3D Gaussian Splatting have improved training efficiency and rendering quality, with extensions like EndoGaussian (Liu et al., 2024b), Endo-GS,

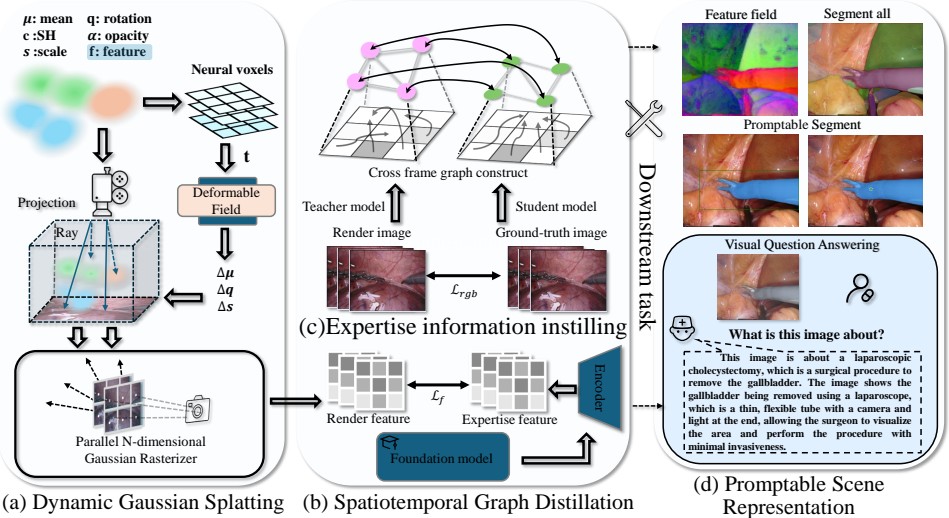

(a) Dynamic Gaussian Splatting  (b) Spatiotemporal Graph Distillation  (d) Promptable Scene Representation

Figure 2: **Overview of the proposed *GaussianClin*.** The framework extends Dynamic Gaussian Splatting by integrating semantic features, enabling versatile downstream real-time clinical tasks such as segmentation, language-guided editing, and medical visual question answering. (a) Dynamic Gaussian Splatting projects neural voxels with deformable fields into a 3D scene. (b) Spatiotemporal Graph Distillation leverages cross-frame information to enhance feature alignment. (c) Expertise information instilling incorporates foundation model priors to improve feature extraction. (d) The final promptable scene representation supports versatile downstream functions.

and DeformGS (Duisterhof et al., 2024) enhancing flexibility and reducing training time. However, medical scene reconstruction remains challenging due to the inherently low surface textures of soft tissues, which provide few reliable visual features for tracking and pose estimation. Additionally, inconsistent lighting conditions in surgical environments, caused by moving instruments and endoscopes, further complicate the process, often requiring the expertise of professionals to correctly interpret and adjust the models.

### 2.3 FEATURE FIELD DISTILLATION FROM FOUNDATION MODELS

Feature field distillation extends 2D visual features into 3D space, enhancing 3D reconstruction. Recent methods, such as NeRF-DFF (Kobayashi et al., 2022), distill 2D feature similarities into 3D, while approaches like SA3D and GaussianGrouping (Ye et al., 2023) refine 3D masks and align 2D masks from foundation models like CLIP (Radford et al., 2021) and SAM (Kirillov et al., 2023) for improved open-vocabulary 3D segmentation. Building on these, our approach enables 3D Gaussian splatting on arbitrary-dimension semantic features via 2D foundation model distillation, improving rendering quality and supporting real-time applications like segmentation, language-guided editing, and medical visual question answering. Prior work, including Semantic NeRF (Zhi et al., 2021) and Panoptic Lifting (Siddiqui et al., 2023), has shown that integrating noisy 2D labels into 3D can yield accurate 3D segmentation. Advancements like Distilled Feature Fields (Shen et al., 2023), LERF (Kerr et al., 2023), and Neural Feature Fusion Fields (Tschernezki et al., 2022) embed pixel-aligned feature vectors for enhanced 3D understanding. However, significant challenges remain in efficiently lifting 2D models into 3D, particularly given limited 3D supervision data and the high computational demands of optimizing neural radiance fields from scratch for each individual scene.

## 3 METHOD

We present *GaussianClin*, an innovative framework that enhances dynamic Gaussian Splatting (GS) for clinical video reconstruction by integrating multimodal feature-based rendering and feature field distillation. Our approach addresses two critical challenges in clinical environments: limited rendering quality due to minimal surface textures and photometric inconsistencies, and the need for advanced semantic understanding. *GaussianClin* embeds trained multimodal feature fields from

pre-trained 2D foundation models into the Gaussian radiance field, enabling the simultaneous representation of both radiance and semantic-rich feature fields. This integration allows for improved performance in various clinical tasks, including 3D tissue visualization, real-time object enhancement, clinical instrument and organ segmentation, and medical visual question answering. The Figure 2 present the architecture of *GaussianClin*.

## 3.1 DYNAMIC GAUSSIAN SPLATTING

Gaussian Splatting (Kerbl et al., 2023) utilizes a set of dense Gaussians to represent 3D data and achieve real-time rendering of scenes. Each Gaussian $\Theta$ is defined by its center $\boldsymbol{\mu} \in \mathbb{R}^3$, covariance matrix $\boldsymbol{\Sigma} \in \mathbb{R}^{3 \times 3}$ (decomposed into a scaling factor $\boldsymbol{s} \in \mathbb{R}^3$ and a rotation quaternion $\boldsymbol{q} \in \mathbb{R}^4$), opacity $\sigma \in \mathbb{R}$, and SH coefficients $\boldsymbol{\alpha} \in \mathbb{R}^C$ for colors and view-dependent appearance. The Gaussian function is expressed as:

$$G(\boldsymbol{x}) = e^{-\frac{1}{2}(\boldsymbol{x}-\boldsymbol{\mu})^T \boldsymbol{\Sigma}^{-1}(\boldsymbol{x}-\boldsymbol{\mu})}, \tag{1}$$

For static scenes, the attributes of the i-th Gaussian are defined as $\Theta_i = x_i, q_i, s_i, \alpha_i, c_i$. The rendering process follows the differentiable 3D Gaussian splatting framework, where the color $C$ of a pixel is computed using volumetric rendering:

$$C = \sum_{i \in \mathcal{N}} c_i \alpha_i \prod_{j=1}^{i-1} (1 - \alpha_j), \tag{2}$$

where $\mathcal{N} = \texttt{overlap}(M, \mu, q, s)$ is the set of 3D Gaussians overlapping the given pixel, determined by the view matrix $M = [R, T]$. To handle dynamic scenes, the framework is extended by integrating temporal information (Wu et al., 2023). A spatial-temporal feature field $\mathcal{E}$ and an MLP decoder $\mathcal{D}$ model the deformation of each Gaussian over time. Given the Gaussian center $\boldsymbol{\mu} = (x, y, z)$ and a query time $t$, the deformation in position, rotation, and scaling is computed, and the Gaussian parameters are then updated:

$$\Delta\boldsymbol{\mu}, \Delta\boldsymbol{q}, \Delta\boldsymbol{s} = \mathcal{D}(\mathcal{E}(\boldsymbol{\mu}, t)), \quad \boldsymbol{\mu}', \boldsymbol{q}', \boldsymbol{s}' = \boldsymbol{\mu} + \Delta\boldsymbol{\mu}, \boldsymbol{q} + \Delta\boldsymbol{q}, \boldsymbol{s} + \Delta\boldsymbol{s}. \tag{3}$$

For dynamic scenes, the rendering process applies volumetric rendering to these dynamically updated Gaussians, allowing for real-time adaptation of the scene representation to account for movements and changes in the environment:

$$C' = \sum_{i \in \mathcal{N}'} c_i \alpha_i \prod_{j=1}^{i-1} (1 - \alpha_j), \tag{4}$$

where $\mathcal{N}' = \texttt{overlap}(M, \mu', q', s')$ is the set of updated dynamic Gaussians overlapping the given pixel.

## 3.2 DYNAMIC FEATURE FIELD FOR 4D GAUSSIANS

Building upon the basic Gaussian Splatting, we incorporate semantic features into the dynamic 3D representation. Each Gaussian $\Theta_i$ now includes a semantic feature vector $\boldsymbol{f}_i \in \mathbb{R}^N$, where $N$ is the latent dimension. These features, derived from 2D foundation models like SAM, CLIP-LSeg, and LLAVA-Med, capture semantic information crucial for tasks such as 3D segmentation, language-guided editing, and medical visual question answering. Given a view matrix $M = [R, T]$, we extend the rendering process to handle high-dimensional feature maps. The semantic feature value $F$ of a pixel is computed using volumetric rendering:

$$F = \sum_{i \in \mathcal{N}} f_i \alpha_i \prod_{j=1}^{i-1} (1 - \alpha_j), \tag{5}$$

where $\mathcal{N} = \texttt{overlap}(M, \mu, q, s)$ is the set of Gaussians overlapping the given pixel. This integration enables simultaneous rendering of RGB images and semantic maps in 3D space, with semantic features optimized alongside geometric and appearance attributes.

**Spatiotemporal Feature Rendering** For dynamic scenes, each Gaussian is extended to include a semantic feature vector $\boldsymbol{f}_i \in \mathbb{R}^N$ and a temporal component $t$, capturing spatial, semantic, and

temporal dynamics. We leverage multiple foundation models such as SAM (Kirillov et al., 2023), LLAVA-Med (Li et al., 2024a), etc. to extract multi-modal features, which are distilled into the dynamic Gaussian representation. This unified spatiotemporal framework enables simultaneous rendering of RGB images, semantic maps, and temporal evolution in dynamic space, enhancing real-time dynamic 3D tasks while maintaining high rendering quality and temporal consistency.

Given the previously introduced view matrix $M = [R, T]$ and a newly incorporated timestamp $t$, our rendering process extends differentiable Gaussian splatting to accommodate both high-dimensional feature maps and temporal dynamics. Our dynamic Gaussian splatting framework transforms the original 3D Gaussians $\mathcal{G}$ into time-dependent 3D Gaussians $\mathcal{G}'(t)$, preserving the efficacy of differential splatting as outlined in (Wang et al., 2019). Upon projecting these dynamic Gaussians onto the 2D image plane, we compute the semantic feature value $F'$ and color $C'$ for each pixel using volumetric rendering, which inherently accounts for temporal variations:

$$C' = \sum_{i \in \mathcal{N}'} c_i \alpha_i \prod_{j=1}^{i-1} (1 - \alpha_j), \quad F' = \sum_{i \in \mathcal{N}'} f_i \alpha_i \prod_{j=1}^{i-1} (1 - \alpha_j), \tag{6}$$

where $\mathcal{N}'$ is the set of 4D Gaussians overlapping the given pixel $\mathcal{N}' = \texttt{overlap}(M, \mu', q', s')$, and $c_i$, $f_i$, and $\alpha_i$ are time-dependent color, feature, and opacity values respectively. Effectively, by conditioning the model on time, we reshape semantic and spatial representations at each frame, enabling real-time handling of dynamic tasks like object tracking, motion segmentation, and temporal language-guided editing. Next, we introduce how the model is supervised by a sequence of "teacher" feature maps $F(t)$ from pre-trained 2D foundation models, allowing it to learn high-level spatiotemporal semantic features. This temporal supervision ensures consistent high-quality rendering across time-varying scenes, accurately capturing both appearance and motion.

### 3.3 Instilling 2D Knowledge via Spatiotemporal Graph Distillation

To address the unique challenges of clinical videos, including rapid camera motion and complex tissue deformations, we introduce a spatiotemporal graph distillation mechanism. This approach is based on the insight that while 2D dynamics in the video might be complex, the underlying 3D motion is often low-dimensional and composed of simpler units of rigid motion.

**Spatiotemporal Feature Graph** Our Spatiotemporal graph distillation process fuses information across frames, creating a globally coherent representation of both scene geometry and motion. We define similarity measures for spatial and temporal components with enhanced expressiveness. At the spatial interaction level, we focus on capturing relationships between different regions within a single frame. The main goal is to identify and fuse information from spatially analogous areas across the image plane, which helps in constructing a coherent global representation of the surgical scene. We define a spatial similarity function based on feature representations of spatial regions. At the temporal coherence level, we extend the graph-guided distillation to capture temporal dynamics between frames. This step ensures that the motion trajectories across frames are aligned, allowing us to leverage temporal consistency to improve the overall representation. The temporal similarity function is defined similarly to the spatial one, but it operates across time, as follows:

$$\texttt{Sim}(s_i, s_j) = [W_s F(s_i)]^T [W_s F(s_j)], \quad G_s(i, j) = \texttt{Top-K}(\texttt{Sim}(s_i, s_j)). \tag{7}$$

Here, $s_i$ and $s_j$ represent spatial regions within a frame, $F(s_i)$ and $F(s_j)$ are their feature embeddings, and $W_s$ is a learnable transformation matrix that projects these features into a shared space. The spatial similarity $Sim(s_i, s_j)$ is computed as the inner product of the transformed features. To construct the spatial graph $G_s$, we retain only the top-K most similar spatial pairs for each region, ensuring that the graph remains sparse and computationally efficient.

To guide the learning process, we introduce a loss function that encourages high similarity between spatially and temporally related regions or frames. Our distillation loss $\mathcal{L}_s$ integrates spatiotemporal similarity losses:

$$\mathcal{L}_s = \sum_{(i,j) \in G_s} (1 - \texttt{Sim}(s_i, s_j))^2 \tag{8}$$

where $\lambda_s$ and $\lambda_t$ are weighting factors that balance the contributions of the spatial and temporal losses. The term $(1 - \texttt{Sim})$ ensures that the loss is minimized when the similarity is close to 1 (i.e.,

perfect similarity), encouraging spatially and temporally related regions or frames to have similar feature embeddings.

By combining these spatial interactions and temporal coherences graphs, our model effectively captures both spatial coherence within frames and temporal consistency across frames, leading to a more robust and holistic understanding of the surgical video. The photometric loss is defined as:

$$\mathcal{L}_{\text{rgb}} = (1 - \lambda)\mathcal{L}_1(I, \hat{I}) + \lambda\mathcal{L}_{\text{D-SSIM}}(I, \hat{I}), \tag{9}$$

combining $\mathcal{L}_1$ loss and structural similarity index (SSIM) to ensure both pixel-level accuracy and perceptual consistency. The feature loss is computed as:

$$\mathcal{L}_f = \|F_t(I) - F_s(\hat{I})\|_1, \tag{10}$$

where $F_t(I)$ is the feature map from a 2D foundation model applied to the ground truth image $I$, and $F_s(\hat{I})$ is the feature map produced by our model for the rendered image $\hat{I}$. Bilinear interpolation is applied to $F_s(\hat{I})$ to ensure matching resolution with $F_t(I)$.

**Overall Optimization** Finally, our *GaussianClin* framework employs a multi-objective optimization strategy to build a robust 4D representation of surgical scenes. The overall loss function combines photometric and semantic consistency:

$$\mathcal{L}_{\text{total}} = \mathcal{L}_{\text{rgb}} + \lambda_f\mathcal{L}_f + \lambda_s\mathcal{L}_s, \tag{11}$$

where $\lambda_f$, $\lambda_s$, and $\lambda_t$ are hyperparameters that balance the contributions of the feature loss, spatial distillation loss, and temporal distillation loss, respectively. These hyperparameters are tuned to optimize the trade-off between photometric accuracy, semantic consistency, and spatiotemporal coherence in the reconstructed dynamic surgical scene representation.

### 3.4 PROMPTABLE SCENE REPRESENTATION WITH VERSATILE FEATURE RENDERING

Our approach leverages foundation models as a base layer of knowledge and capabilities adaptable to various tasks. Through our feature field distillation technique, we create practical 3D representations from these foundation models' features. We focus primarily on the Segment Anything Model (SAM) (Kirillov et al., 2023) and its variants, including SAM2 and Grounded SAM, as well as LLaVA-Med for medical visual question answering. These models enable zero-shot or few-shot capabilities in various medical imaging tasks without requiring task-specific training. Our teacher-student distillation framework extends these 2D capabilities—prompted by points, boxes, or text—into 3D space, creating refined feature fields that preserve the versatility of the original models.

Our method for promptable explicit scene representation operates as follows: Given a 3D Gaussian $x$ from a set of $N$ ordered Gaussians that overlap with a target pixel, denoted as $x_i \in \mathcal{X}$ where $\mathcal{X} = \{x_1, \ldots, x_N\}$, we compute the activation score of a prompt $\tau$ on the 3D Gaussian $x$ using the cosine similarity between the query $q(\tau)$ in the feature space and the semantic feature $f(x)$ of the Gaussian. The similarity score $s$ is given by:

$$s = \frac{f(x) \cdot q(\tau)}{||f(x)|| \cdot ||q(\tau)||}, \tag{12}$$

If we have a set $\mathcal{T}$ of potential labels, such as a text label set for semantic segmentation or a point set representing possible pixels for a point-prompt, the probability of a prompt $\tau$ corresponding to the 3D Gaussian $x$ is obtained via a softmax function:

$$\mathbf{p}(\tau|x) = \text{softmax}(s) = \frac{\exp(s)}{\sum_{s_j \in \mathcal{T}} \exp(s_j)}. \tag{13}$$

We leverage these computed probabilities to filter out Gaussians with low probability scores. This selective filtering enables various operations, such as extraction, deletion, or appearance modification, by updating the color $c(x)$ and opacity $\alpha(x)$ values accordingly. With the updated color set $\{c_i\}_{i=1}^n$ and opacity set $\{\alpha_i\}_{i=1}^n$, where $n$ is smaller than $N$, point-based $\alpha$-blending can be applied to render the edited radiance field from any novel viewpoint. This framework for promptable explicit scene representation facilitates several advanced functionalities.

By integrating advanced 2D models, our framework enables real-time, zero-shot semantic segmentation without task-specific training. This approach excels in complex surgical environments, rapidly

Table 1: **Novel view rendering results** on EndoNeRF (Wang et al., 2022), Endovis17 (Allan et al., 2019) and Endovis18 (Allan et al., 2020) datasets.

| Metrics | EndoNERF_cutting | | | EndoNERF_pulling | | | Endovis17 | | | Endovis18 | | |
|---|---|---|---|---|---|---|---|---|---|---|---|---|
| | PSNR ↑ | SSIM ↑ | LPIPS ↓ | PSNR ↑ | SSIM ↑ | LPIPS ↓ | PSNR↑ | SSIM ↑ | LPIPS ↓ | PSNR↑ | SSIM ↑ | LPIPS ↓ |
| Base 3DNERF | 23.02 | 0.7930 | 0.3531 | 22.09 | 0.8106 | 0.3878 | 10.53 | 0.5448 | 0.4569 | 18.26 | 0.7716 | 0.3530 |
| Base 4DNERF | 31.35 | 0.8913 | 0.1327 | 28.90 | 0.8573 | 0.1704 | 17.41 | 0.7472 | **0.3515** | 18.89 | 0.8028 | 0.3562 |
| NeRF-DFF | 23.27 | 0.7117 | 0.4559 | 27.45 | 0.8326 | 0.2383 | 16.14 | 0.6464 | 0.3567 | 19.08 | 0.7857 | **0.3379** |
| Base 3DGS | 22.79 | 0.7889 | 0.3857 | 21.03 | 0.8027 | 0.4313 | 18.74 | 0.7913 | 0.4125 | 18.93 | 0.7965 | 0.5231 |
| Base 4DGS | 34.10 | 0.9299 | 0.1209 | 28.94 | 0.8847 | 0.1433 | 19.12 | 0.7947 | 0.4028 | 19.06 | 0.8002 | 0.5173 |
| Feature DS | 22.84 | 0.7909 | 0.3811 | 21.07 | 0.8027 | 0.4241 | 19.21 | 0.7959 | 0.4087 | 19.13 | 0.7981 | 0.5198 |
| *GaussianClin* | **35.31** | **0.9424** | **0.0928** | **29.41** | **0.8887** | **0.0945** | **20.26** | **0.8205** | 0.3649 | **19.69** | **0.8120** | 0.4849 |

identifying diverse tissues, instruments, and anatomical structures, even those unseen during training. The system supports language-guided editing, allowing surgeons to modify 3D scenes with natural language commands, enhancing surgical planning and guidance. Furthermore, Med-LLM integration enables medical visual question answering, allowing professionals to query the 3D scene using natural language. These features advance real-time decision-making and precision in surgeries, making the dynamic Gaussian splatting framework highly effective for medical applications. Additional task instructions are detailed in Appendix B.

# 4 EXPERIMENTS

## 4.1 DATASETS AND EVALUATION

Our evaluation utilizes four diverse datasets: EndoNeRF, EndoVis18, EndoVis17, and EndoVis Conversations, each presenting unique challenges in surgical scene understanding. These datasets cover in-vivo prostatectomy, robotic instrument segmentation, depth estimation, tracking, and visual question-answering tasks, providing a robust foundation for assessing our method across various surgical contexts. We employ comprehensive metrics to evaluate performance, including PSNR, SSIM, and LPIPS for image quality; GPT-4 Score, accuracy, F-score, and mIoU for semantic understanding; and training time, inference speed, and GPU memory usage for efficiency. This holistic approach enables thorough comparisons with state-of-the-art methods and identifies areas for improvement in our 4D surgical scene reconstruction framework. Detailed implementation information is available in Appendix A.

## 4.2 IMPROVED RENDERING QUALITY

Our method leverages semantic features to empower models to comprehend unseen labels by mapping semantically close medical concepts to similar regions in the embedding space. This advancement notably promotes scalability in information acquisition and scene understanding, facilitating a profound comprehension of intricate surgical scenes. We distill multimodal features for this novel view semantic segmentation task in surgical environments. Our experiments demonstrate the improvement of incorporating semantic features over naive 3D and 4D Gaussian Splatting methods. As shown in Table 1, our *GaussianClin* surpasses baseline 3D and 4D Gaussian models in performance metrics across all datasets: EndoNeRF_cutting, EndoNeRF_pulling, Endovis17, and Endovis18.

In our comparison with NeRF-DFF using these datasets, we address the potential trade-off between the quality of the semantic feature map and RGB images. Our model demonstrates higher accuracy across all metrics, including PSNR, SSIM, and LPIPS. For instance, on the EndoNeRF_cutting dataset, our method achieves a PSNR of 35.31, SSIM of 0.9424, and LPIPS of 0.0928, significantly outperforming NeRF-DFF and other baselines. Notably, our approach yields better visual quality on novel views and semantic segmentation masks for both synthetic and real surgical scenes compared to NeRF-DFF and other baseline methods. This improvement in rendering quality, coupled with the enhanced semantic understanding, paves the way for more accurate and detailed surgical scene reconstruction and analysis.

Table 2: **Segmentation results (Ground SAM)** for rendered images from novel viewpoints.

| Metrics | EndoNERF_cutting | | | EndoNERF_pulling | | | Endovis17 | | |
|---|---|---|---|---|---|---|---|---|---|
| | IOU↑ | DICE ↑ | FPS↑ | IOU↑ | DICE ↑ | FPS ↑ | IOU ↑ | DICE ↑ | FPS↑ |
| NeRF-DFF | 0.8258 | 0.0218 | 41 | 0.1543 | 0.0050 | 52 | 0.1961 | 0.1256 | 35 |
| FeatureDS | 0.9668 | 0.8993 | 98 | 0.9136 | 0.5422 | 85 | 0.5732 | 0.6644 | 110 |
| *GaussianClin* | **0.9945** | **0.9613** | **125** | **0.9449** | **0.9483** | **138** | **0.9963** | **0.7573** | **115** |

NeRF-DFF
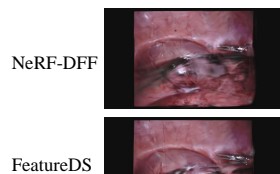

FeatureDS

Ours
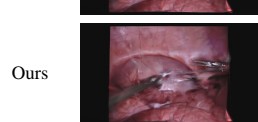

Rendered novel views

**Q1: What type of procedure is being depicted in this image?**
**A1:**
**NeRF-DFF:** This appears to be an open heart surgery.
**FeatureDS:** This looks like a laparoscopic gallbladder removal.
**Ours:** The image shows a robot-assisted surgery being performed on kidney.
**GT:** The image shows a robot-assisted minimally invasive surgery being performed on isolated porcine kidney.

**Q2: What is the state of the prograsp forceps?**
**A2:**
**NeRF-DFF:** The prograsp forceps are closed and not in use.
**FeatureDS:** The prograsp forceps are cutting tissue.
**Ours:** The ProGrasp forceps are being used to retract tissue.
**GT:** The prograsp forcep is engaged in retraction.

Figure 3: **Quantitative results of Medical Visual Question Answering (VQA)**: Compared to NeRF-DFF and Feature DS, our approach yields precise and comprehensive responses.

## 4.3 MEDICAL VISUAL QUESTION ANSWERING

As shown in Fig 3, our method addresses the limited class diversity in medical datasets by leveraging advanced semantic features to map unseen labels to similar embedding regions, enhancing scalability and understanding of complex surgical scenes. Results in Tab. 3 show significant improvements in both segmentation accuracy and rendering speed compared to existing methods like NeRF-DFF and FeatureDS across multiple datasets. For example, on EndoNERF_cutting, our method achieves an IOU of 0.9945 and DICE score of 0.9613, far surpassing NeRF-DFF.

## 4.4 NOVEL VIEW SEMANTIC SEGMENTATION

Our method addresses the limitation of fewer classes in medical datasets by leveraging advanced semantic features, enabling models to comprehend unseen labels by mapping semantically close medical concepts to similar regions in the embedding space. This promotes scalability in information acquisition and facilitates a profound comprehension of intricate surgical scenes, ultimately enhancing the model's adaptability to diverse medical scenarios.

We distill multimodal features for novel view semantic segmentation tasks in surgical environments. As shown in Table 2, our *GaussianClin* substantially outperforms NeRF-DFF and FeatureDS across all datasets, demonstrating significantly higher accuracy in IOU and DICE scores. Notably, our approach achieves superior rendering speeds, more than doubling the frame rate per second compared to NeRF-DFF while maintaining state-of-the-art segmentation performance. This improvement in both segmentation accuracy and rendering speed, along with our method's ability to generalize across different surgical scenarios, demonstrates its potential for enhancing various aspects of computer-assisted surgery, including real-time guidance, automated surgical workflow analysis, and advanced training simulations.

## 4.5 LANGUAGE-GUIDED EDITING

Figure 4 showcases our novel view editing results, demonstrating successful extraction and deletion of surgical instruments based on language inputs. Our approach provides cleaner extractions with minimal artifacts. The model's ability to selectively edit specific elements while preserving surrounding structures demonstrates its potential for enhancing surgical visualization and decision-making. This functionality opens new possibilities for interactive surgical planning, intraoperative

Table 3: **Visual Question Answering (VQA) results by distilling Med-LLaVa**: We use the GPT-4 score to measure the answering performance and FPS for efficiency.

| Metrics | Endovis17-C | | Endovis18-C | |
|---|---|---|---|---|
| | GPT-4 score ↑ | FPS↑ | GPT-4 score↑ | FPS ↑ |
| NeRF-DFF | 76.89 | 42 | 74.57 | 69 |
| Feature DS | 80.12 | 88 | 78.15 | 78 |
| *GaussianClin* | **82.97** | **123** | **81.25** | **114** |

Table 4: Ablation study for key components of *GaussianClin* on EndoNeRF .

| Metrics | Image | | | Feature |
|---|---|---|---|---|
| | PSNR↑ | SSIM↑ | LPIPS↓ | PSNR↑ |
| No Dynamics | 22.84 | 0.7909 | 0.3811 | 23.89 |
| No Feature | 34.05 | 0.9269 | 0.1209 | - |
| No Graph | 34.65 | 0.9336 | 0.1002 | 27.46 |
| Full Model | **35.31** | **0.9424** | **0.0928** | **28.65** |

guidance, and advanced medical education tools, leveraging a comprehensive understanding of 3D surgical environments from any viewpoint.

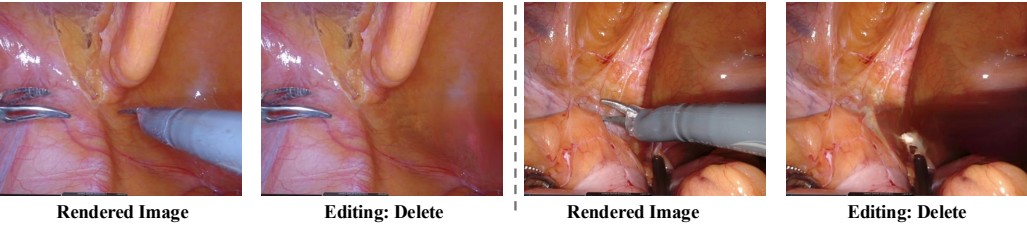

**Rendered Image**        **Editing: Delete**        **Rendered Image**        **Editing: Delete**

Figure 4: **Editing Results on EndoNeRF** : With the text of *"Delete tools"*, indicating our method's ability to accurately remove surgical instruments from rendered feature.

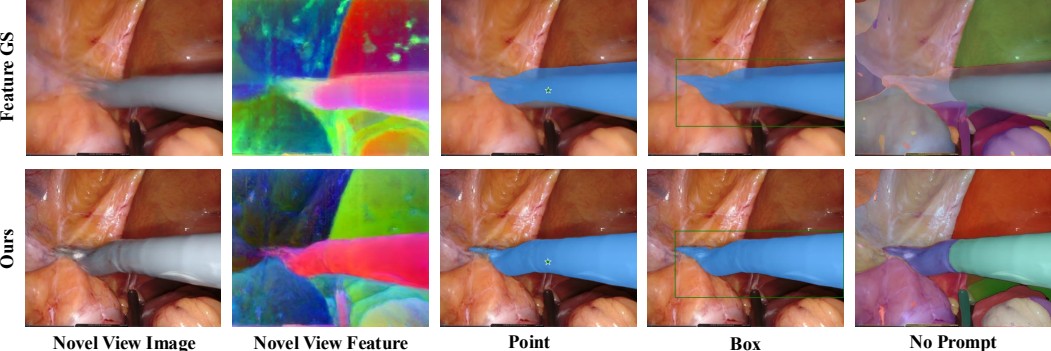

**Novel View Image**    **Novel View Feature**    **Point**    **Box**    **No Prompt**

Figure 5: **Comparison of Novel View Segmentation results with Feature GS**: Feature GS approach exhibits lower reconstruction quality and less precise segmentation masks. Our method achieves higher-quality masks, providing more detailed rendering.

### 4.6 PROMPTABLE SEMANTIC SEGMENTATION FROM ANY VIEW

We conduct promptable semantic segmentation from novel views on the EndoNeRF (Wang et al., 2022). Experimental results are shown in Fig 5. we use PCA-based feature visualization (Pedregosa et al., 2011) to demonstrate that our high-quality segmentation masks result from superior feature rendering. We can observe that in comparison to Feature GS, *GaussianClin* exhibits superior rendering quality, enhanced segmentation detail, and an improved feature map.

## 5 CONCLUSION

In this work, we introduced *GaussianClin*, an innovative approach that addresses the challenges of reconstructing dynamic 3D models in clinical environments. By leveraging multimodal feature-based Gaussian splatting and incorporating spatiotemporal graph distillation, *GaussianClin* effectively captures complex tissue deformations and enhances tasks such as 3D tissue visualization and medical instrument segmentation. The integration of general medical knowledge from pre-trained foundation models further improves performance in real-time applications like medical visual question answering. Our experimental results demonstrate the robustness and practicality of *Gaussian-Clin* in clinical settings, highlighting its potential to significantly advance robot-assisted surgery.

## ETHIC STATEMENT

This paper does not raise any ethical concerns. This study does not involve any human subjects, practices to data set releases, potentially harmful insights, methodologies and applications, potential conflicts of interest and sponsorship, discrimination/bias/fairness concerns, privacy and security issues, legal compliance, and research integrity issues.

## REPRODUCIBILITY STATEMENT

To make all experiments reproducible, we have listed all detailed hyper-parameters of our methods. Due to privacy concerns, we will upload the anonymous link of source codes and instructions during the discussion phase to make it only visible to reviewers.

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

## A  IMPLEMENTATION DETAILS

Our implementation is tailored to address the specific challenges presented by each dataset. For the EndoNeRF dataset, which features complex tissue deformations and tool occlusions, we initialize by randomly sampling 0.1% of points to reduce redundancy. We employ Adam as the optimizer with an initial learning rate of $1.6 \times 10^{-3}$. Our training strategy involves a warmup phase to optimize Canonical Gaussians for 1k iterations, followed by optimization of the entire framework for 3k iterations. For the EndoVis18 and EndoVis17 datasets, which focus on instrument segmentation

and tracking, we adapt our model to handle the temporal aspects of the data. We split the frame data of each scene into 7:1 training and testing sets. For the EndoVis Conversations dataset, which incorporates visual question-answering tasks, we set the epoch to 20, batch size to 16, and learning rate to $1 \times 10^{-5}$. This setting allows our model to effectively learn the relationship between visual features and textual queries. All experiments are conducted on a single RTX 4090 GPU. The training process for each dataset takes approximately 2 hours. We also evaluate a high-quality version in our experiments, where we optimize all basis parameters for each Gaussian attribute.

## B  DOWNSTREAM TASKS FOR VERIFYING THE SCALABILITY AND WIDE APPLICABILITY OF *GaussianClin*

**Real-time Semantic Segmentation** By seamlessly integrating advanced foundation model priors such as Segment Anything Model (SAM) and CLIP-LSeg into our innovative 4D feature fields, we have unlocked the capability for real-time, zero-shot semantic segmentation. This groundbreaking approach transcends traditional segmentation methods by eliminating the need for task-specific training or predefined categories. Instead, it offers a flexible, domain-agnostic segmentation framework that can adapt to a wide array of scenarios using open-set text labels or feature queries. This versatility is particularly crucial in complex surgical environments, where the diversity of tissue types, instruments, and anatomical structures poses significant challenges for conventional segmentation techniques. Our method's ability to rapidly identify and delineate various elements within a surgical scene, even those not encountered during training, represents a significant leap forward in intraoperative imaging and computer-assisted surgery. By providing instant, accurate segmentation of both familiar and novel objects, our system enhances surgical precision, facilitates real-time decision-making, and potentially improves patient outcomes across a broad spectrum of surgical procedures.

**Language-guided Editing** Our framework revolutionizes the interaction with 3D surgical scenes by introducing intuitive, language-guided editing capabilities. This advanced functionality allows users to manipulate complex 3D environments using natural language commands, bridging the gap between human intent and computational action. For instance, surgeons or medical professionals can input directives such as "Remove the surgical tool on the right," "Highlight the tumor area," or "Isolate the blood vessels near the incision site." The system interprets these natural language inputs and translates them into precise, context-aware modifications of the 3D scene. This capability is particularly transformative for surgical planning and training scenarios, where it enables rapid prototyping of surgical approaches and interactive exploration of anatomical structures. By allowing users to dynamically alter the virtual surgical environment through simple verbal or textual commands, our system enhances the potential for highly realistic and responsive surgical simulations. This not only accelerates the learning curve for trainee surgeons but also provides experienced practitioners with a powerful tool for preoperative planning and intraoperative guidance. Moreover, the language-guided editing feature opens up new possibilities for collaborative surgical planning, enabling multiple specialists to easily communicate and visualize complex procedural steps, thereby fostering more comprehensive and efficient surgical strategies.

**Medical Visual Question Answering (VQA)** Our approach extends beyond static scene understanding to enable dynamic interaction through medical VQA, allowing medical professionals to query the 3D reconstructed surgical scene using natural language questions and receive informative responses based on the scene's content and dynamics. This capability combines 4D semantic features with advanced natural language processing techniques, encoding questions and matching them against relevant features in the scene. By enabling these advanced semantic interactions, our method bridges the gap between 3D reconstruction and high-level understanding, paving the way for more intelligent and interactive surgical assistance systems. This promptable explicit scene representation opens new possibilities for human-computer interaction in medical contexts, supporting tasks from surgical planning to intraoperative guidance and post-operative analysis, thereby enhancing the overall efficacy and utility of our 4D Gaussian splatting framework in medical applications.

