# OpenReview forum: "GaussianClin: Multimodal Featured Gaussian Splatting for Dynamic Clinical Videos"
_ICLR.cc/2025/Conference — ICLR 2025 Conference Withdrawn Submission_

### Official Review · Reviewer_iLwT · 2024-10-23

**Soundness:** 3
**Presentation:** 3
**Contribution:** 3
**Rating:** 3
**Confidence:** 5

**Summary:**

The author proposed a multimodal featured Gaussian Splatting GaussianClin method for the clinic dynamic 3D scene reconstruction. Compared to the existing GS in the surgical domain, they author 1) include the features from most recent foundation model to improve the surgical scene understanding. 2) The author proposed spatialtemporal feature graph to matain the coherent representation of geometric and motion in the surgical video. 3) The system support various downstream tasks such as language-guided editing and medical VQA.

**Strengths:**

1. The authors' motivation is clear—they aim to enhance the performance of dynamic surgical scene reconstruction and support various downstream tasks.
2. The text and figures clearly explain the methods, and the related work is sufficiently covered.
3. Based on tod shows ihe design, the proposed methmproved performance across different tasks.

**Weaknesses:**

1. In line 131, the author claims one contribution is accelerating training and inference speed, but the manuscript does not report the training times for the proposed method or comparison methods.
2. In line 727, the author mentions the language-guided editing function can **"highlight the tumor area"** and **"isolate blood vessels near the incision site,"** but no results for these functions are shown.
3. In line 296, the author discusses the challenge of **rapid camera motion**, yet the three datasets used for experiments have a fixed camera position.
4. In line 323, the parameters $\lambda_s$ and $\lambda_t$ are not clearly defined or used in the manuscript.
5. The language-guided tool deletion is unnecessary since the datasets already contain segmentation masks of tools, which can be used to filter out regions in the NeRF and GS settings.
6. There is no reference provided for the Feature DS method.
7. The manuscript lacks comparisons to recent methods like EndoGaussian [1], and the reconstruction quality appears lower than these methods.

[1] Liu, Yifan, et al. "Endogaussian: Gaussian splatting for deformable surgical scene reconstruction." arXiv preprint arXiv:2401.12561 (2024).

**Questions:**

See the weaknesses.

---

### Official Review · Reviewer_iuWj · 2024-10-28

**Soundness:** 3
**Presentation:** 2
**Contribution:** 3
**Rating:** 5
**Confidence:** 2

**Summary:**

The paper is positioned in the area of dynamic scene reconstruction/rendering from medical videos (e.g., endoscopy videos) and proposes a new method that extends 3D Gaussian splatting in two directions: (1) Integration of multi-modal features maps derived from foundation models to improve down stream tasks such as tissue/instrument segmentation or VQA. (2) A spatio-temporal graph that improves spatial and temporal regularity of the representations. Experiments on standard endoscopy benchmark datasets that cover a wide range of downstream tasks showcase the applicability of the method, which quantitatively outperforms the tested baselines across nearly all standard metrics.

**Strengths:**

- I believe that paper tackles a highly relevant problem with a high significance and while the paper is (to some degree at least) a combination existing techniques, it is nicely executed and presents in total a new approach.
- Interesting idea to incorporate a range of foundation models to inject clinical knowledge into the modelling process (the coining "clinical knowledge" might be challenged, though; see below).
- The variety of potential downstream tasks (6 in total) that are tackled by the paper is quite interesting and showcase the general applicability of the proposed method.

**Weaknesses:**

- The paper is extremely busy and it is hard to assess what the exact methodological contributions are. What parts of the spatio-temporal graph distillation method are really novel, for example? How does the feature distillation method presented here differ methodologically from the other approaches mentioned in 2.3? Is the innovation mainly the application to a medical setup? In my opinion, it would have been better to substantially shorten the first two section of the paper to have additional space to make the contributions more explicit in Sec. 3.
- The paper claims that the use of the foundation models equips the model with clinical knowledge. I am not entirely sure if that's true and if it is not more an incorporation of general object detection etc. skills that help to better segmentation structures. I am especially surprised about this claim as the majority of foundation models included are general purpose model such as SAM.
- I find the chosen baselines in parts of the experiments quite unconvincing. For example, in Sec. 2.2 several recent 4D approaches are discussed (e.g, EndoGaussian), but none of them is actually used in the experiments. While I recognize that they may not be usable for all downstream tasks, wouldn't it have been possible to use them at least for the novel view rendering part? On the other hand, the chosen baseline FeatureDS does not seem to be described at all. In general, a better justification of the baselines for each experiment is critically needed.
- I am also struggling with fully understanding the training process in the way the paper is written. Are all feature maps derived from all possible foundation models used simultaneously? Is everything trained end-to-end?
- A major contribution is the spatio-temporal regularization approach, but I fail to see a specific experiment that evaluate this aspect. Where can I find results that show its effectiveness?

**Questions:**

Please see my weaknesses, which already contain questions for the authors. Most crucial are those related to the methodological novelty and the baselines.

---

### Official Review · Reviewer_ztyt · 2024-11-03

**Soundness:** 3
**Presentation:** 2
**Contribution:** 3
**Rating:** 5
**Confidence:** 3

**Summary:**

The paper presents a Gaussian Splatting-based model designed for rendering surgical scenes, with two main contributions: the incorporation of foundation model embeddings into the Gaussian representation and the integration of a spatiotemporal distillation graph. The foundation model embeddings enable prompt-based processing, facilitating prompt-driven view synthesis within the rendering workflow. Experimental results indicate that the embedding information further enhances rendering quality.

**Strengths:**

The paper presents an interesting concept that extends the Gaussian representation with additional image-based feature embeddings, enabling additional applications like segmentation, captioning, and other downstream applications.

Results suggest that the additional features contribute to the rendering quality of the model.

**Weaknesses:**

Additional clarity regarding the addition of the feature embeddings into the Gaussian model can be suggested. Similarly, it is recommended to give some details regarding the training/validation/testing setup used during the experiments (See questions)

**Questions:**

Regarding the clarity of the model's description, the Gaussian includes an additional feature representation in their definitions. While it is clear that the features $F$ for a particular pixel are generated by blending the embeddings $f$ of each overlapping Gaussian, I would like to ask for additional comments on:
* How are the features incorporated into each individual Gaussian? - is this feature obtained from pre-trained foundation models and then assigned to the individual Gaussians? In general, how are these Gaussian-level embedding features initialized?
* Related to the previous question, I understand there is a teacher approach that guides the integration process of the embedded features. Then, I understand each individual Gaussian feature $f$ adapts during the learning process. In this regard, how is $f$ represented and updated?
* In section 3.4, a query $q(t)$ is mentioned. How is this query obtained? Does it come from the teacher Foundation model? If this is the case, then is it still necessary to evaluate the foundation model during the inference process? In this case, what would be the advantage of having the feature integrated over the Gaussian vs. evaluating the foundation model on the rendered view?

Some questions regarding the performance of the model:
* How are the training validation and testing views selected?
* Is there any insight regarding the performance of the Gaussian model in recovering the 3D structure of the anatomy?

---

### Official Review · Reviewer_csG5 · 2024-11-04

**Soundness:** 3
**Presentation:** 2
**Contribution:** 3
**Rating:** 6
**Confidence:** 4

**Summary:**

In this paper, the authors are trying to address the complex problem of reconstructing dynamic 3D models from clinical videos. as such tasks are crucial for surgical visualization, robot-assisted surgery, and medical training. The paper shows how clinical environments pose unique challenges such as limited surface textures of tissues, inconsistent lighting conditions, and the need for expert-level medical knowledge to interpret intricate anatomical structures. These factors make it difficult for existing 3D reconstruction techniques to be directly applied effectively in medical settings.To overcome these challenges, the authors present this work which is claimed to be a novel framework that enhances 3D modeling capabilities. Gaussian Splatting is used which is a technique that represents 3D scenes and allows efficient and high-quality rendering. The key innovation in their work is the integration of trained multimodal feature fields from pre-trained 2D foundation models (SAM) into the 3D Gaussian radiance field.

**Strengths:**

This paper addresses the challenging task of reconstructing dynamic 3D models from clinical videos, which is crucial for surgical applications. The proposed framework seems to integrate multimodal features from pre-trained 2D foundation models. I can see that it is aiming to enhance 3D modelling capabilities in medical imaging.

1. **Originality:** Integrating multimodal features into Gaussian Splatting for dynamic 3D reconstruction in a medical context is a novel idea. Intro of a spatiotemporal graph distillation mechanism to handle tissue deformations adds to the originality as well. However, the approach builds upon existing methods, and the degree of innovation might be **_considered incremental rather than groundbreaking._**

2. **Quality:** The paper presents experimental results demonstrating improvements over baseline methods in rendering quality and tasks like segmentation and visual question answering. _While these results are promising, the evaluation lacks depth._ One suggestion here would be to present a more comprehensive comparison with a broader set of metrics would strengthen the validity of the claims.

3. **Clarity:** The paper is generally well-structured, but certain sections suffer from insufficient detail, particularly the explanation of the spatiotemporal graph distillation. _However, I understand the space constraint issues that might have arisen here._

4. **Significance:** I think this work indeed does have potential significance for medical imaging and robot-assisted surgery.

**Weaknesses:**

Some weaknesses/areas of improvement:

1. **Incremental Novelty:** While integrating multimodal features into Gaussian Splatting for dynamic 3D reconstruction is an interesting approach, as I mentioned before, it may be considered an incremental advancement rather than a groundbreaking innovation. The method builds upon existing techniques by combining pre-trained 2D models with 3D reconstruction, which, although useful, might not represent a significant leap in the field.

2. **Limited Experimental Evaluation:** The experimental results are promising but somewhat limited in scope. The evaluations are conducted on a narrow set of datasets, and comparisons are primarily made against baseline methods without including a broader range of SOTA techniques.

3. **Practical Applicability Concerns:** I wonder the extent to which this could be implemented practically. No inclusion of experimenting with real-time performance in clinical settings.

4. **Insufficient Detail on Key Components:** As I told before, the spatiotemporal graph distillation mechanism is a central part of the proposed approach, but the paper provides insufficient detail about its implementation and impact.

5. **Overreliance on Pre-trained Models:** The approach depends heavily on pre-trained 2D models that may not be specifically tailored to medical imaging. This reliance could limit the method's ability to capture specialized anatomical details or pathological variations critical for medical applications, potentially affecting accuracy and reliability. Maybe a fine-tuned model could be a better fit?

**Questions:**

1. The spatiotemporal graph distillation mechanism is a central component of your approach, yet the paper provides limited details about its implementation and impact. Could you elaborate on how this mechanism is constructed and integrated into the model? Have you conducted ablation studies to quantify its contribution to overall performance, like in handling tissue deformations?

2. Given the reliance on large pre-trained 2D foundation models, there may be concerns about the feasibility of deploying GaussianClin in real-time clinical settings where computational resources are limited. Have you assessed the computational requirements and latency of your method in such environments? Can you provide insights into how GaussianClin performs on hardware typically available in clinical settings?

3. One of your stated goals is to address challenges like limited surface textures and inconsistent lighting in surgical environments. However, the paper lacks quantitative analyses demonstrating improvements in these areas. Could you provide specific experiments or metrics that showcase how GaussianClin effectively handles these challenges compared to existing methods?

**Details Of Ethics Concerns:**

No review required

---

### Note · Authors · 2024-11-23

I have read and agree with the venue's withdrawal policy on behalf of myself and my co-authors.